# Contextualizing involvement in terrorist violence by considering non-significant findings: Using null results and temporal perspectives to better understand radicalization outcomes

**Bart Schuurman** *, **Sarah L. Carthy**

Institute of Security and Global Affairs, Leiden University, The Hague, The Netherlands

* b.w.schuurman@fgga.leidenuniv.nl

## Abstract

Irrespective of discipline, the publication of null or non-significant findings is rare in the social sciences. For burgeoning fields like terrorism research, this is particularly problematic. As well as increasing the likelihood of Type II errors, the selective reporting of significant findings ultimately impedes progression, hindering comprehensive syntheses of evidence and enabling ill-supported lines of scientific enquiry to persist. This manuscript discusses several structural and individual-level variables which failed to produce significant, linear associations with involvement in terrorist violence in a dataset ($N = 206$) of right-wing and jihadist extremists active in Europe and North America. After considering methodological factors such as non-random distributions of missing data, we illustrate how certain variables are significantly associated with involvement in terrorist violence at particular periods in a radicalizing individual's lifespan, but not others (i.e., pre- or post-radicalization onset). Moreover, we demonstrate that while some *static*, binary constructs (such as whether or not a radicalizing individual was exposed to diverse viewpoints) are not associated with terrorist violence, their influence *over time* produces different associations. We conclude that radicalization may be less about individuals having pre-disposing risk factors, such as biographical stressors, and more about cognitive changes that allow individuals to re-evaluate their lives through the lens of an extremist ideology. We also underline the importance of taking a temporal, rather than static, perspective to better understand the variables associated with the outcomes of radicalization trajectories.

## Introduction

There is no scarcity of hypotheses for how and why people become involved in terrorism [1–4]. Yet the quantity of such theorizing has frequently been contrasted with concerns over quality. One of the critiques that followed the rapid expansion of research on terrorism after the 9/

**Data Availability Statement:** The dataset, codebook and methodological supplements are available at https://doi.org/10.7910/DVN/NJX5BV.

**Funding:** This research was supported by grant VI. Veni.191R.007 from the Dutch Research Council (https://www.nwo.nl, to BS), grant CRF 8000-21053 from Public Safety Canada (https://www.publicsafety.gc.ca/cnt/bt/cc/fnd-en.aspx, to BS), and subsidy 3985535 from the Dutch Ministry of Justice and Security (https://www.nctv.nl/onderwerpen/bewaken-en-beveiligen). The funders had no role in study design, data collection and analysis, decision to publish or preparation of the manuscript.

**Competing interests:** The authors have declared that no competing interests exist.

11 attacks was that the majority of explanations for terrorism, as well as for the related concept of radicalization, lacked sufficient empirical support [5–7]. Over the past decade, the quality of research on terrorism has improved [8], underpinned by a keen focus on empiricism [9, 10]. Nonetheless, concerns prevail over the robustness of the evidence base for the various factors associated with radicalization and terrorism [11–13]. This manuscript contributes to the field's empirical turn but does so in a somewhat unorthodox fashion. Rather than reporting upon empirically supported hypotheses, we highlight and discuss literature-derived explanations for involvement in terrorist violence for which we *did not* identify statistically significant associations.

Reporting non-significant findings runs counter to a tendency among journals and authors to prefer positive results [14, 15]. Indeed, as noted by Krueger in their appraisal of significance testing more broadly [16], if luck were to grant success with a probability of .05, accepting or rejecting the null hypothesis based on $p < .05$ no longer becomes a matter of significance. Yet, discussing when and why particular analyses do *not* yield the expected outcomes is crucial for several reasons. First, across the social scientific disciplines there has been increasing concern about the poor replicability of many findings [17]. Overcoming this "replication crisis" [18] by publishing null as well as significant findings is likely to lead to a more nuanced understanding of the particular settings or circumstances in which certain variables do or do not exert a statistically significant influence [19].

Second, reporting null findings addresses the "Achilles' heel" [20] of systematic reviews and meta-analyses: publication bias. The disinclination of journals and authors to publish non-significant finding, as well as selective outcome reporting, have been identified as threats to the overall validity of attempts to aggregate and synthesize findings on a particular topic [21]. Reporting where radicalized individuals who have become involved in terrorist violence do *not* differ from their non-involved in terrorist violence counterparts (i.e., where the null hypothesis is supported), accompanies a field-wide drive towards synthesizing the breadth and depth of terrorism and radicalization research [10, 22–25].

Third, the subject of terrorism appears vulnerable to well-entrenched "common sense" assumptions about the relationship between putative risk factors and terrorism which, upon closer examination, have proven erroneous [26] or, at the very least, non-linear [27]. Scholarship on terrorism is also characterized by an emphasis on the "latest threat", limiting the time and attention devoted to in-depth analysis of underlying processes [28]. Reporting statistically non-significant findings can help address both concerns; enabling critical reflection on commonsensical or intuitively appealing explanations that lack statistically significant support, while lending priority to the type of fundamental research questions that often fall by the wayside in the desire to keep abreast of recent developments. What ultimately emerges from reporting non-significant results is a mechanism for theory building through falsification. An aversion to null results underpins the endurance of "unkillable" theories [29, p. 559], whereas theory falsification, principally, allows theories to become more complex by incentivizing the development of more testable theoretical assumptions [30].

This manuscript utilizes the "(Non-) Involvement in Terrorist Violence" (NITV) dataset [31]. This dataset was constructed to ascertain the differences between radicalized individuals who engage in terrorist attacks, and radicalized individuals who do not. While the dataset supports the examination of linear associations, it is not the intention of the authors to reduce the complexity of radicalization trajectories to isolated, "causal" factors. Instead, the 159 variables in the dataset reflect a range of theories and insights related to (non-) involvement in terrorist violence. These are intended to allow for the initial saliency of certain variables in the radicalized sub-populations (i.e., those involved and non-involved in terrorist violence) to be examined, supporting future lines of academic enquiry.

Our manuscript begins by providing a brief overview of the project's theoretical underpinnings. We then outline our methodological approach, contribution to the literature, and limitations, before highlighting those variables which we did not find to be significantly associated with involvement in terrorist violence at the bivariate level ($p > .05$). Wherever feasible, our dataset records whether a variable exerted an influence before as well as after radicalization onset. This allows us to not just report non-significant findings, but, for some of them at least, to assess changes in significance level over time. We use this temporal perspective, as well as broader debates on involvement in terrorist violence, to contextualize our results in the discussion section.

## Researching involvement in terrorist violence

The NITV dataset has its origins in an extensive review of the literature on (non-) involvement in terrorist violence. The theories and explanations that emerged from this review were operationalized as variables in our codebook, enabling exploratory and inductive research on why some people who radicalize become involved in terrorist violence, while others do not. As the non-significant findings reported on in this manuscript are drawn from this review, a brief summary is provided of the levels of analysis that we utilized, and the main insights that each provides into (non-) involvement in terrorist violence. In addition to the terrorism-focused literature, we also drew from disciplines such as criminology and psychology to inform the development of our *structural*, *group* and *individual* level variables.

Structural-level explanations for terrorism consider the characteristics of the broader social and political environment in which terrorism emerges. While generally considered insufficient as explanations for individual or group involvement in political violence, structural-level factors can enable or constrain opportunities for such violence to emerge [4, 32, 33]. Considerable work has been done on exploring the mobilizing potential of grievances related to socioeconomic hardship [34], poor education [35], and lack of political representation [36]. There is also a rich tradition of scholarship looking at the role of (perceived) state repression, and the ways in which increasingly antagonistic interactions between states and non-state challengers, such as protest movements, can contribute to the outbreak of political violence [37–40]. Furthermore, scholars have taken an interest in transnational "spillover" effects, noting that conflict in one particular country or region can affect a geographically distant locale through diaspora links, or because overseas military interventions create an impetus for retribution [41, 42].

The use of structural-level variables in our project was subject to several qualifications. Notably, because of limited empirical support for a linear relationship between socioeconomic hardship and involvement in terrorism [27, 43–46], and because our cases are all drawn from advanced and at least relatively prosperous Western states, we did not consider poverty a relevant structural-level factor (we did consider socioeconomic stressors at the individual-level of analysis). Our project's specific geographical focus on Europe and North-America also meant that structural-level theories developed in relation to other parts of the world, such as those that focus on the disruptive effects of modernization [47] or democratization [48], were not considered. Instead, the emphasis of our structural-level variables lay on assessing whether antagonistic and violent interactions between states and non-state challengers, as well as (perceived) lack of political representation, could contribute to a radicalizing's individual likelihood of engaging in terrorist violence.

As pointed out, structural-level factors are important but ultimately insufficient explanations for terrorism as there are always many more people exposed to them than the small number that comes to engage in political violence. Social psychologists and social movement

researchers have long demonstrated the importance of also seeing radicalization as an interactive process in which small-group dynamics, the characteristics of broader social movements, and the responses offered by state and society to the claims that such movements put forward, play crucial roles in influencing participants' views and behaviors [49–54]. Exposure to others who justify or even use terrorism (i.e., social learning theory) is particularly salient in this regard [55–57], especially so when those others include "role models" whose charisma or perceived authority make them particularly likely to be emulated [58]. Peer pressures are also relevant here, as the desire for peer-approval, or fear of expressing dissent, can propel individuals to become involved in extremist movements in the first place [59–62], and, once involved, can influence their adoption or maintenance of extremist views [2, 63].

Extremist groups and movements also offer a variety of benefits important for understanding why individuals become and then stay involved in such entities, despite the considerable risks of prosecution or death that they face for doing so [61, 64, 65]. These benefits are partly instrumental; groups simply have more resources to effectively organize for violence than most individuals [50, 51], making them attractive as means for redressing the grievances of individual participants [66]. But extremist groups can also offer a set of emotional appeals that are key in binding participants to their cause and each other. These include a strong sense of purpose and meaning through adherence to a cause seen to be of grave or even transcendent importance, as well as comradeship and belonging found among an inner-circle of similarly committed individuals [64, 67–69]. Beyond such social-psychological aspects, scholars have also highlighted that certain characteristics of extremist groups can affect their likelihood of engaging in ideologically-motivated violence. These include their size and age, whether or not members have (para)military experience, as well as (changing) internal norms on the moral appropriateness and strategic utility of violence [70–74].

Finally, a significant body of work has taken the radicalizing individual as its object of study. These efforts have been partly conceptual; clarifying that radicalization trajectories have cognitive and behavioral dimensions, that the former spans a spectrum of ideological commitment, that the latter similarly encompasses a broad behavioral repertoire, and that cognitive radicalization does not necessarily lead to behavioral radicalization [75–78]. Essentially, people can adopt radicalized worldviews without acting upon them. However, when they do, such behavioral radicalization can include a range of both legal and illegal activities beyond terrorist violence, such as spreading propaganda, raising funds, recruiting new members, or vying for political power through democratic means [79–81]. In other words, although radicalization has become almost synonymous with terrorism, radicalized behavior can take a variety of forms, including both essentially legal and non-violent activism, as well as clearly illegal political violence [82, 83].

Alongside such conceptual work, empirical research has asked whether particular characteristics, dispositions or experiences can explain an individual's susceptibility to radicalization, or the particular outcomes of such trajectories [10, 84]. Some of these explanations have incorporated established theories from adjacent disciplines; social control theory [84, 85], for instance, posits that an absence of pro-social ties such as employment, education and family commitments increases the likelihood that an individual will engage in delinquent behavior, including radicalization and terrorism. Others have begun to adopt life-course perspectives similar to those used in criminology [86], examining whether a range of factors related to childhood victimization or parenting styles are relevant to understanding radicalization onset later in life [87]. The pre-radicalization period has also been studied from the perspective of criminal antecedents, hypothesized to increase the likelihood of future delinquency, including extremism-related crimes [11, 88]. Particular attention has been paid to the potential influence of non-clinical and clinical predispositions such as self-control and mental illness, with recent

years seeing the emergence of a more nuanced and empirical understanding on this subject than was prevalent in previous decades [24, 89].

Because radicalization is a process resulting in the perception that violence is a necessary instrument to advance or safeguard cherished values or political ambitions, many accounts of terrorism have focused on *how* such cognitive changes take place [90]. This has resulted in a variety of risk factors for terrorism such as perceived in-group threat [91], perceived disadvantage vis-à-vis peers or other societal groups [92], vicarious suffering through identification with victims of perceived injustice [93], the dehumanization of opponents [94], the role of emotions such as anger [66], and the gradual loss of trust in democratic politics and institutions [95]. The role of extremist beliefs themselves has also drawn significant attention [96], with authors considering whether different degrees of commitment to these worldviews can account for the variety of violent and non-violent behaviors exhibited by radicalized individuals [97]. A desire for personal significance, potentially as a way of alleviating mortality salience [98], has been at the core of another strand of research on the causes of terrorism [67]. Individual-focused accounts have also yielded a large number of more descriptive insights, for instance on how gender, age, access to weapons and training in their use can impact radicalization onset and outcomes [57, 99].

## Research design

This brief overview illustrates the main analytical perspectives that we used to study (non-)involvement in terrorist violence (the NITV dataset itself can be found online [31], see also S3 File). From these insights, we created a 159-variable codebook to systematically collect data on individuals who had radicalized to extremism in Europe and North America. We use this specific phrase to emphasize that our dataset distinguishes between radicals and *extremists*, focusing exclusively on the latter. While radicals seek far-reaching political or societal change, they tend to do so within the boundaries of the existing order (i.e., usually without wanting revolutionary change) and generally without recourse to violent means. Extremists, on the other hand, hold the use of violence to be singularly effective. They are not content with piecemeal changes to a system they consider rotten and, instead, seek revolutionary upheaval [100]. Of these two cognitive frames, only extremism can be readily tied to the use of political violence in general and terrorism in particular.

Cognizant that the definition of terrorism remains subject to debate [101], we opted for Schmid's [102] "Academic Consensus" definition, which is not only based on input by a range of experts, but also strives for greater objectivity than is common in many "official" definitions of terrorism through its focus on terrorism as a behavior, rather than a form of violence specific to a type of (non-state) actor. Hence, we defined terrorism as the premeditated (threatened) use of deadly violence against civilians or non-combatants, intended principally as a means of violent communication, enabling perpetrators to draw attention to their cause, coerce opponents and inspire potential adherents.

The NITV dataset encompasses 206 individuals who radicalized to extremism, half of whom were involved in the planning, preparation or perpetration of terrorist violence ($n = 103$) with the remainder operationalized as non-involved in terrorist violence ($n = 103$). This enabled us to comparatively assess whether the various theories and findings outlined in the literature review were relevant for understanding different radicalization outcomes. Conversely, this approach also allowed us to consider *negative* results, identifying factors which were not significantly associated with the outcome. Where possible, we ascertained whether variables were present "before" and (or) "during" radicalization, allowing us to establish whether a temporal relation between variables existed [103].

## Sampling

Three case selection criteria were used. First, to ensure broadly comparable socio-cultural backgrounds, only extremists who, through citizenship or upbringing, had strong ties of belonging to the countries in which they radicalized were selected for inclusion [104]. As such, foreign nationals who traveled to Europe or North America solely to conduct a terrorist attack there, were excluded. Second, we excluded individuals who had joined overseas insurgent groups, such as the large number of Salafi-Jihadists who traveled to Syria and Iraq to fight on behalf of organizations such as the so-called Islamic State. While these "foreign fighters" have certainly used terrorist tactics, we consider insurgent warfare in the Middle East to be strategically and contextually distinct from terrorism in Western states, so much so that its inclusion could have unduly blurred the boundaries of the phenomenon being studied [105]. Finally, to ensure that our results would be relevant for understanding the two forms of extremism most likely to contribute to terrorist violence in Western states [106, 107], we focused exclusively on Salafi-Jihadist ($n$ = 103) and right-wing ($n$ = 103) extremists.

The inclusion of both right-wing and jihadist extremists, as well as group- and lone-actor terrorists, in our dataset may raise concerns over sample biases that deserve some clarification. Research has shown that ideology influences a variety of aspects related to terrorist groups and movements, such as their lethality [108, 109] and odds of organizational survival [110]. Differences have also been demonstrated at the individual level of analysis, with research suggesting, for instance, that right-wing terrorists in 1960's West-Germany were more socioeconomically deprived, and educationally underperforming, than their left-wing counterparts [111]. Similar contrasts emerged in research on U.S.-based terrorists active in the same time period [112, 113]. More recent work has suggested that jihadists tend to be older, are more likely to have military experience and less likely to have criminal records than right-wing extremists [114]. On the whole, however, scholars appear to have assumed that individual-level predictors of involvement in terrorism are at least relatively similar [115, 116]. This appears to be the case especially when it comes to the broader characteristics of radicalization processes, for instance in terms of the roles that grievances, social learning and perceptions of in-group threat (among many other factors) play [38, 57, 117].

Accordingly, we have worked from the assumption that while the predictors of involvement in jihadism and right-wing extremist terrorism are unlikely to be the same, they are likely to be similar enough to warrant both types of individuals being included in our dataset. Similarly, while lone actors do show distinctiveness on a number of predictors, perhaps especially so with regard to higher rates of diagnosed mental illness [24, 118], such unique attributes appear to be outweighed by similarities with their group-based counterparts [119–121]. Hence, we argue that it makes sense to include both lone actors and group-based extremists in our work. It should also be noted that we did not specifically select cases or controls based on whether they were lone actors or operated in a group context. Instead, we specifically examined group membership as a variable of interest.

Our sampling approach began by considering all terrorist attackers (i.e., our cases) active in North-America and Europe since the 1980s. This "total population sampling" approach is common in comparative research with small populations or infrequent phenomena [122 p. 3], allowing for better generalizability [123] and reducing risk of bias [24]. We then identified individuals who had cognitively and behaviorally radicalized without planning, preparing or engaging in terrorist violence (i.e., our controls).

The identification of suitable cases relied primarily on publicly available sources. These included the academic literature on terrorism and radicalization, court records, overviews of terrorist attacks in Europe and North America, and datasets such as the Global Terrorism

**Table 1. Countries in the NITV dataset.**

| Country | Number of individuals | Percentage of total |
|---|---|---|
| United States | 79 | 38.3 |
| Germany | 35 | 17.0 |
| The Netherlands | 25 | 12.1 |
| United Kingdom | 25 | 12.1 |
| Sweden | 11 | 5.3 |
| France | 9 | 4.4 |
| Canada | 8 | 3.9 |
| Belgium | 4 | 1.9 |
| Norway | 4 | 1.9 |
| Austria | 2 | 1.0 |
| Denmark | 2 | 1.0 |
| Australia | 1 | 0.5 |
| Switzerland | 1 | 0.5 |
| *Total* | **206** | **100.0** |

Database (GTD). We also drew on specialist collections of primary data on terrorists and extremists, such as the one maintained by Oslo University's Centre for Research on Extremism (C-REX), reports by government agencies such as EUROPOL, and numerous journalistic accounts. Additionally, we utilized our professional networks and snowball-sampling related to the interviews that we conducted (see below), to identify additional relevant cases. When selecting individuals for our non-involved in terrorist violence sub-sample, we prioritized individuals who had left extremism behind as a safeguard against selecting persons who had simply not been able to conduct an attack before they were arrested.

Not all of the circa 260 cases that we identified were included in the final sample. Some individuals who were described, either in the available sources or by themselves, as extremists were, upon closer inspection, better defined as radicals, and the information pertaining to others was simply too sparse to allow for effective coding. The 206 cases included in the NITV dataset represent a broad range of countries (Table 1).

To avoid sampling bias, we sought to match "involved" and "non-involved" sub-groups according to extremist affiliation. Although there were equal numbers of right-wing extremists and jihadist extremists in the sample, due to data-related limitations, we found it challenging to identify, in particular, jihadist *controls* and right-wing *cases*. As such, the matching procedure was not 1:1. Some right-wing cases had more than one control, while a number of jihadist cases lacked individual controls (Table 2).

## Data collection & coding

We ascertained exposure by using a codebook of largely binary items (i.e., the presence or absence of phenomena). All cases were coded by the authors between January 2020 and

**Table 2. Cases and controls distributed according to ideological conviction.**

| | Extremist conviction | | |
|---|---|---|---|
| **Outcome** | **Right-wing extremist** | **Jihadist extremist** | |
| Involved in terrorist violence | *n* = 44 | *n* = 59 | *n* = **103** |
| Non-involved in terrorist violence | *n* = 59 | *n* = 44 | *n* = **103** |
| | | | |
| | *n* = **103** | *n* = **103** | |

December 2021, using primary data wherever possible. This consisted of semi-structured interviews with former extremists and terrorists ($n = 37$), autobiographical materials written by extremists and terrorists ($n = 56$) and case files provided by the Dutch Public Prosecution Service ($n = 19$). The identification of potential interviewees followed from the general case selection guidelines outlined above. Only "homegrown" extremists of either jihadist or right-wing extremist convictions were considered and, for the non-involved controls, their participation in extremism must have ended to ensure that they were not just "non-involved" because arrest cut short their ability to engage in an attack. Initial contact was made through a variety of channels, including e-mails or direct messages to individuals with an online presence, e-mails to publishers to request their help as intermediaries in reaching extremists who had written autobiographies, e-mails or phone calls to former extremists whom we had encountered as part of previous research projects, and snowball-sampling where appropriate to reach potential interviewees through individuals we had already spoken with (i.e., gatekeepers). In some cases, the interview formed the primary evidentiary basis whereas in others it was used to supplement other data sources. This meant that the specific topics covered differed from interview to interview, depending on whether they were intended to cover all or some of the items listed in our codebook (See S1 and S2 Files for further details).

Because radicalized individuals are a small and hard to reach segment of the population, we also utilized secondary sources such as academic publications, court verdicts and journalistic accounts, triangulating data where possible to minimize biases. Interrater-reliability coding (IRR) took place over an 8-month period to ensure uniformity in coding decisions. As defined by Landis and Koch [124], our average IRR-score indicated "substantial" agreement levels with the role of chance accounted for through Cohen's kappa calculations ($M = .65$, $SD = 0.06$). IRR scores never dropped below "moderate" levels of agreement. Ethics approval for the project was granted in November 2019 by our faculty ethics board (reference: 2019-012-ISGA-Schuurman). All data were anonymized before being recorded and interviews were conducted on the basis of informed consent (see also S2 File).

## Statistical procedure

We utilized SPSS 25 [125] to examine associations between the variables in our codebook and the outcome of interest (i.e., involvement in terrorist violence). We interpreted tests of significance at or below $p = .05$ as significant. In line with Visentin, Cleary and Hunt [126], before concluding that an association does or does not exist, we first considered whether positive or negative results may reveal a Type I or Type II error as a result of missing data weakening statistical power.

The influence of missing data on the reliability of research findings has long been a challenge in epidemiological work, including studies on terrorism and radicalization [127]. When distributed randomly, missing data are not usually an impediment to obtaining unbiased results [128]. However, when data is missing non-randomly, the likelihood that confounding influences are undermining the reliability of the results increases markedly. Consequently, to more confidently conclude that non-significant findings represented "no association" rather than a Type II error owing to missing data, all associations reported in this manuscript underwent a standardized procedure to ascertain whether data were missing at random. Following recommendations set out by Perkins et al. [129], valid and invalid cases were compared across age, gender, outcome and, for specific variables, what type of source was utilized (e.g., semi-structured interview or secondary literature). Where confounding was identified, the variable was excluded from prevalence reporting, or carefully caveated. Details of our missing data procedure are available as an online supplement (S1 Table) with missing data percentages provided in the Results section.

## Contribution to the literature

To the best of our knowledge, this manuscript is the first of its kind to specifically utilize null results to quantitatively explore why only some individuals who radicalize become involved in terrorist violence. However, we are by no means the first to consider broader questions surrounding why, among a group of people who share particular (extremist) convictions, only some are willing to engage in risky or outright violent behavior in pursuit of those shared ideals. For instance, "differential recruitment" [130, p. 787], or why some people, but not others ostensibly similar to them, are drawn to particular social movements [66, 130–132] is a well-established line of inquiry. Its emphasis on relational dynamics has inspired a rich literature in which the emergence of political violence and terrorism have been studied from the perspective of group dynamics, as well as the increasingly antagonistic relationships between social movements and their opponents [37, 38, 40, 93, 133]. From this perspective, the strength and extent of social ties to likeminded peers and others already involved in high-risk activism, or outright political violence, are vital in shaping the likelihood that an individual will, or will not, move from ideological affinity with an extremist cause to actual participation [49].

The terrorism-focused literature that emerged after 9/11 has had a strong emphasis on the individual level of analysis, rather than groups or broader social movements [10, 49, 68]. In this body of work, it has become well-established that there is no linear relationship between cognitive and behavioral radicalization, nor any one particular way in which such behavioral radicalization can manifest itself [75, 97, 134–136]. In other words, it has long been clear that radicalizing influences, regardless of whether they reside at the structural, group or movement, or individual level of analysis, do not produce uniform results among people exposed to them. Recent years have also seen scholars increasingly try to differentiate between various forms of radicalization process outcomes (e.g., violent versus non-violent) [57, 83, 84, 99, 120, 137, 138], as well as specific attempts to better understand why radicalizing individuals may *not* turn to violence [139–141]. Clearly, some ground has already been covered on the question of why (a particular form of) radicalization *does not* occur, bringing with it a better sense of the contexts in which theories for involvement in terrorism do, and do not, apply.

This literature provides multiple starting points for investigating when, why, or how radicalization trajectories do, or do not, lead to involvement in terrorist violence. As far as we are aware, however, our study is the first to specifically focus on statistically non-significant findings to gain a better understanding of these questions. Our goal here is not simply to label theories that seem of little relevance for understanding involvement in terrorist attacks (such generalizations would in any case be beyond the confines of our research design), Instead, we use non-significant findings as a starting point for considering the broader circumstances that influence whether a particular variable is significantly associated with this outcome. We do so, in part, by contrasting a particular theory or assumption with our actual findings. However, because we assessed the influence of our predictors both before and after radicalization onset wherever feasible, we are also able to assess potential changes in a variable's level of significance over time. Combined with a specific focus on both risk and protective factors for involvement in terrorist violence, the latter of which are still infrequently used in the overwhelmingly risk-focused terrorism literature [11, 142–144], we are able to offer nuanced and multifaceted insights into (non-) predictors of involvement in terrorist violence unique to this field of study.

Furthermore, at $N = 206$, we are able to draw on a sizeable dataset that is larger than several other recent additions to this literature [99, 120, 138], although it remains smaller than LaFree et al.'s pathbreaking work on the distinction between violent and non-violent radicalization outcomes [84]. By including cases from a range of European and North American countries

(Table 1) we also offer broader geographical coverage than extant terrorism-focused datasets, which tend to focus specifically on English-speaking countries [84, 120, 137]. Moreover, by including right-wing extremist as well as Salafi-Jihadist individuals in our dataset, our results are relevant to the extremist currents most likely to pose a persistent terrorist threat to (Western) states.

Finally, although not formally designed as such, our project can also be seen as a type of replication study. By highlighting those variables for which we did not find a statistically significant relationship with our outcome of interest, we are able to reflect on the validity of underlying theoretical assumptions. The topic of replication, or the reproducibility of a particular study's results, has in recent years attracted considerable interest [145]. Particularly as scholars across a variety of disciplines, including psychology [18], medicine [146], and (to a lesser degree) economics [147], noted with alarm that the results of many studies failed to replicate. Whereas a 2016 survey by the journal *Nature* revealed that over 70% of scientists polled had engaged in replication studies [145], a 2018 survey among 75 terrorism researchers revealed the opposite; most had not tried to reproduce previous studies' results [148]. Indeed, two long-time scholars of terrorism have noted the "continuing dearth of evaluation and replication studies" [149, p. 68].

Replication studies are not entirely absent from terrorism research. They have been used, *inter alia*, in research on the effect of troop deployments on transnational terrorism [150], the relationship between homicide and terrorism at a national level [151], individuals' sympathy for violent protest and terrorism [152], the relationship between media coverage of terrorism and the occurrence of attacks [153], the willingness of "gatekeepers" to offer help to individuals displaying problematic (violent) behavior [154], and the public's psychological reaction to terrorist attacks [155]. Still, the overall picture is one of a field that has -so far- spent relatively little effort on replication work, in the process lagging behind more established disciplines that have given this subject considerable attention over the past several years. We can only hypothesize about why this is so. Terrorism studies has long suffered from issues related to data and methods [8, 156–158], of which the limited attention given to replication may be another facet. It could also reflect that this is still a field in relative infancy, where, important exceptions notwithstanding [84, 159], the emphasis remains on descriptive work and theory development, rather than theory testing [6, 11]. Whatever the case may be, we believe that our results also contribute to strengthening research on terrorism through a greater emphasis on replicability.

## Limitations

Our work is subject to several limitations that need to be clearly acknowledged for our results to be accurately conveyed. The first is that we realize that caution should be used in treating *p*-values in a dichotomous fashion, rather than as a continuous measure in which the -somewhat arbitrary- cut-off point for significance is usually placed at .05 [160]. Just because a variable was found to have a *p*-value higher than .05 does not necessarily mean that there is no association [161]. In the same way that statistical significance is not equivalent to theoretical or clinical relevance or significance [162], we do not wish to dismiss the relevance of particular variables to theoretical conceptualizations of involvement in terrorist violence. Instead, we consider the context in which these null findings emerge, reflect on their temporal relations and draw particular attention to where the replication of previous findings has not occurred. In doing so, we hope to contribute to the creation of the necessary conditions for the development of more complex, foundational theories of (non-) involvement in terrorist violence.

Second, we are cognizant that our approach to null-hypothesis testing, while observant of potential contributors to Type II errors, falls prey to the same methodological drawbacks were

**Table 3. Sources used for data collection.**

| Source Type | Frequency | % |
|---|---|---|
| Secondary only | 117 | 56.8 |
| Semi-structured interviews | 37 | 18.0 |
| Auto-biographical materials | 56 | 27.2 |
| Police investigative files[a] | 19 | 9.2 |

*Notes. N* = 206. Several cases drew on numerous data sources, causing frequency totals to exceed the number of cases in our dataset, and percentage totals to exceed 100.

[a] Provided by the Dutch Public Prosecution Service.

we to produce significant findings. While our "total population sampling" approach [122] allowed for better generalizability [123], demonstrating a *non*-observation is an inherently challenging pursuit, particularly if we are to consider the complexity and inter-relatedness of constructs measured in the social sciences [163]. Relatedly, an indicator of maturity in any given field exploring causal hypotheses is the emergence of mediators and moderators in standard, bivariate relations [164], and a key limitation of the current research is its antecedent position in this regard.

Third, although we sought to match cases and controls according to ideological conviction, we could not do so while prioritizing high quality data. That being said, even if we had been able to produce a 1:1 matched sample, it would still have been vulnerable to the same drawbacks as most matching based on demographic variables, whereby one case could be matched to other controls without substantially changing the association [165]. Thus, we wish to emphasize that our sample is best described as loosely-matched rather than individually matched.

Fourth, it is important to emphasize that we did not consider it appropriate to use psychometric measures to code data drawn from the sources available to us (Table 3). As a result, our measurement instrument for certain, individual-level variables was subject to coder biases. For instance, with regard to grievances, we used the primary and secondary sources available to us to determine whether there were tangible feelings of injustice emerging from comparisons that the radicalized individual made with other individuals or societal groups at the political, religious, or personal levels (see also S3 File). These different types of grievances were measured in a binary fashion before being combined into a measure of the presence or absence of grievances more broadly. While we were sensitive to the relative nature of the phenomenon (considering, for instance, that two people in comparable contexts may perceive their situation in different ways [166]), this approach likely failed to capture the full complexity of comparisons at the cognitive level [167]. This limitation stems from the broader challenge of accessing these hard-to-reach populations [8, 168, 169], making the implementation of standardized measures impractical. It is for this reason that we present and interpret our findings, first and foremost, at the exploratory level.

Fifth, and relatedly, research on terrorism has frequently been criticized for an overreliance on secondary sources, particularly media reporting [8, 158, 170]. While journalists often provide invaluable descriptions of terrorist plots and their perpetrators, research has also shown that such accounts are prone to factual errors, contradictory findings, or editorial biases that lead to uneven coverage of extremism-related events [171, 172]. Except, perhaps, for the deadliest terrorist attacks, media coverage is also likely to be limited in the scope and detail of its coverage. Reliance on secondary data is therefore likely to yield inaccuracies, inconsistencies or missing data [173].

To compensate for these issues, we tried to draw on primary data sources wherever possible. These principally consisted of semi-structured interviews with former extremists and

terrorists, autobiographical materials, and police investigative files provided by the Dutch Public Prosecution Service (Table 3). While such primary data tends to be considerably more detailed than what can be garnered from secondary sources, it is certainly not without its limitations [173]. Interviewees may be unreliable, for instance, because of a desire to portray their past conduct in a more favorable light, or they may simply be unable to accurately recall events that transpired years ago [168]. The material in police files is collated specifically to enable criminal prosecution, rather than objective academic analysis, which adds a notable bias to such sources [174]. Autobiographical materials, especially those related to something as controversial as terrorism, may be marred by self-aggrandizement, exaggeration, omission of incriminating details, or be purposefully misleading [175, 176]. More generally, the privileged nature of information drawn from primary data often makes it difficult for other researchers to falsify claims made on the basis of such evidence [177].

These issues cannot be fully overcome. However, we sought to address them as best we could by using multiple sources, across a range of source types, wherever possible. For instance, by drawing on an autobiographical account but combining and critically appraising that information with insights provided by journalists, as well as sentencing information. Such combinations of primary and secondary data is not uncommon, especially in "nascent" areas of research [178, p. 105], and all the more so where hard-to-reach populations, such as radicalized individuals, are concerned [179–181]. Finally, the various issues surrounding source-reliability notwithstanding, our extensive use of primary data remains noteworthy in a field in which such material has historically been scarce [8, 22, 158, 182].

## Results

Table 4 lists all variables from the NITV codebook which, despite having random distributions of missing data and equal distribution between groups, did not emerge as statistically significant predictors of involvement in terrorist violence. In line with the guidelines for reporting non-significant statistical results set out by Visentin, Cleary and Hunt [126], the observed differences (i.e., the effect sizes) are reported along with the $p$-value and confidence intervals. For ease of reading, variables are grouped according to the level of analysis used, with the individual-level variables given a further subdivision that mirrors our codebook (see also S3 File). Where we ascertained the presence of a particular variable both before and during radicalization, both $p$-values and effect sizes are provided. Variables with a pre- and post-radicalization onset component were only included in our results if distributions of missing data were random across all variables. As can be seen, only the structural and individual-level variables provided null-results that met this criterion.

## Discussion

On our outcome of interest, involvement in terrorist violence, our analyses revealed dozens of null findings, prompting two broad lines of discussion. First, we consider the disparity between particular null findings and what others have postulated or identified in the broader terrorism literature. Second, we consider disparities in associations identified between comparable variables measured at different temporal stages (i.e., before or during radicalization). We discuss our findings in the order that they appear in Table 4.

### Structural-level findings

Although previous research has indicated that overseas armed conflicts can "spillover" into domestic terrorism, for instance across diaspora links [186], we failed to find a statistically significant association between the presence of such foreign conflicts and the likelihood that a

**Table 4. Non-significant structural and individual-level associations.**

| Variable | Operationalization | Involved in terrorist violence | | | Non-involved in terrorist violence | | | p | OR | 95% CI | % Missing data |
|---|---|---|---|---|---|---|---|---|---|---|---|
| | | Yes | No | Sum | Yes | No | Sum | | | | |
| *Structural level of analysis* | | | | | | | | | | | |
| Conflict "spillover" influenced radicalization onset | No / Yes / Unknown | 53 (54.1) | 45 (42.5) | 98 | 42 (40.8) | 61 (59.2) | 103 | .06 | 1.71 | 0.98–2.99 | 2.4 |
| Presence of a "foreign fighter" locale[a] | No / Yes / Unknown | 4 (9.1) | 40 (90.9) | 44 | 3 (5.1) | 56 (94.9) | 59 | .43 | 1.87 | 0.40–8.80 | <1.0 |
| *Individual level of analysis: biographical details* | | | | | | | | | | | |
| Tertiary education completed prior to radicalization | No / Yes / Unknown | 14 (14.0) | 86 (86.0) | 100 | 16 (15.5) | 87 (84.5) | 103 | .76 | 0.89 | 0.41–1.92 | 1.5 |
| Educational enrolment during radicalization | No / Yes / Unknown | 61 (59.2) | 42 (40.8) | 103 | 68 (66.0) | 35 (34.0) | 103 | .31 | 0.75 | 0.42–1.32 | 0.0 |
| Education abandonment during radicalization[b] | -2 Abandoned; -1 Less time; 0 Stability; 1 More time | **-2** 28 (49.1) **-1** 6 (10.5) | **0** 23 (40.4) **1** 0 (0.0) | 57 | **-2** 21 (31.8) **-1** 7 (10.6) | **0** 31 (47.0) **1** 7 (10.6) | 66 | .05 | 2.07 | 0.99–4.30 | 5.0 |
| Employed or in school prior to radicalization | No / Yes / Unknown | 88 (86.3) | 14 (13.7) | 102 | 95 (93.1) | 7 (6.9) | 102 | .11 | 0.46 | 0.18–1.20 | 1.0 |
| Employed or in school during radicalization | No / Yes / Unknown | 73 (71.6) | 29 (28.4) | 102 | 87 (86.1) | 14 (13.9) | 101 | .01 | 0.41 | 0.20–0.82 | 1.5 |
| Diagnosed neurodevelopmental disorder | No / Yes / Unknown | 12 (13.0) | 80 (87.0) | 92 | 12 (12.9) | 81 (87.1) | 93 | .98 | 1.01 | 0.43–2.39 | 10.2 |
| Substance abuse before radicalization | No / Yes / Unknown | 27 (28.1) | 69 (71.9) | 96 | 23 (23.0) | 77 (77.0) | 100 | .41 | 1.31 | 0.69–2.50 | 4.9 |
| Substance abuse during radicalization | No / Yes / Unknown | 21 (20.8) | 80 (79.2) | 101 | 28 (27.7) | 73 (72.3) | 101 | .25 | 0.68 | 0.36–1.31 | 1.9 |
| Non-violent criminal antecedents | No / Yes / Unknown | 45 (45.5) | 54 (54.5) | 99 | 22 (22.0) | 78 (78.0) | 100 | <.00 | 2.96 | 1.59–5.47 | 3.4 |
| Non-violent crime during radicalization | No / Yes / Unknown | 57 (55.9) | 45 (44.1) | 102 | 50 (50.0) | 50 (50.0) | 100 | .40 | 1.27 | 0.72–2.20 | 1.9 |
| Violent criminal antecedents | No / Yes / Unknown | 30 (29.4) | 72 (70.6) | 102 | 15 (14.7) | 87 (85.3) | 102 | .01 | 2.42 | 1.20–4.84 | 1.0 |
| Violent crime during radicalization | No / Yes / Unknown | 36 (36.0) | 64 (64.0) | 100 | 33 (32.4) | 69 (67.6) | 102 | .59 | 1.18 | 0.66–2.11 | 1.9[1] |
| Criminal gang membership | No / Yes / Unknown | 12 (11.9) | 89 (88.1) | 101 | 5 (4.9) | 97 (95.1) | 102 | .07 | 2.62 | 0.88–7.72 | 1.5[1] |
| Relationship before radicalization | No / Yes / Unknown | 26 (27.7) | 68 (72.3) | 94 | 27 (27.6) | 71 (72.4) | 98 | .99 | 1.01 | 0.54–1.89 | 6.8 |
| Relationship during radicalization | No / Yes / Unknown | 46 (45.1) | 56 (54.9) | 102 | 65 (63.1) | 38 (36.9) | 103 | .01 | 0.48 | 0.27–0.84 | <1.0 |
| Relationship breakup during radicalization[c] | Break up / Stability | 15 (32.6) | 31 (67.4) | 46 | 16 (24.6) | 49 (75.4) | 65 | .36 | 1.48 | 0.64–3.42 | 0.0 |
| Children born before radicalization | No / Yes / Unknown | 9 (8.7) | 94 (91.3) | 103 | 13 (12.7) | 89 (87.3) | 102 | .35 | 0.66 | 0.27–1.61 | <1.0 |
| Children born during radicalization | No / Yes / Unknown | 15 (14.6) | 88 (85.4) | 103 | 35 (34.3) | 67 (65.7) | 102 | .00 | 0.33 | 0.16–0.65 | <1.0 |
| Homeless before radicalization | No / Yes / Unknown | 5 (4.9) | 97 (95.1) | 102 | 5 (4.9) | 98 (95.1) | 103 | .99 | 1.01 | 0.28–3.60 | <1.0[1] |
| Homeless during radicalization | No / Yes / Unknown | 5 (4.9) | 97 (95.1) | 102 | 9 (8.8) | 93 (91.2) | 102 | .27 | 0.53 | 0.17–1.64 | 1.0 |

*(Continued)*

**Table 4.** (Continued)

| Variable | Operationalization | Involved in terrorist violence | | | Non-involved in terrorist violence | | | | | | |
|---|---|---|---|---|---|---|---|---|---|---|---|
| | | **Yes** | **No** | **Sum** | **Yes** | **No** | **Sum** | **p** | **OR** | **95% CI** | **% Missing data** |
| Convert before radicalization | No / Yes / Unknown | 43 (43.0) | 57 (57.0) | 100 | 36 (35.6) | 65 (64.4) | 101 | .29 | 1.36 | 0.77–2.40 | 2.4 |
| Convert during radicalization | No / Yes / Unknown | 5 (4.9) | 97 (95.1) | 102 | 9 (8.7) | 94 (91.3) | 103 | .28 | 0.54 | 0.17–1.67 | <1.0 |
| | | **M** | **SD** | **n** | **M** | **SD** | **n** | **p** | **d** | **95% CI** | **% Missing data** |
| Total time radicalized (months) | Continuous | 91.9 | 106.3 | 101 | 104.8 | 92.0 | 102 | .35 | -0.13 | -0.40–0.14 | 1.5 |
| Social isolation before radicalization | 5-point scale [183] | 2.4 | 1.05 | 100 | 2.3 | 1.07 | 102 | .61 | -0.08 | -0.35–0.20 | 1.9 |
| Social isolation during radicalization[d] | | 2.6 | 1.13 | 103 | 2.0 | 0.84 | 102 | .00 | -0.87 | -1.15--0.58 | <1% |
| *Individual level of analysis: family & upbringing* | | | | | | | | | | | |
| Number of siblings | Continuous | 3.3 | 1.82 | 99 | 2.9 | 1.66 | 99 | .07 | 0.23 | -0.05–0.51 | 3.9 |
| | | **Yes** | **No** | **Sum** | **Yes** | **No** | **Sum** | **p** | **Odds Ratio** | **95% CI** | **% Missing data** |
| Family involved in extremism/terrorism | No / Yes / Unknown | 7 (6.9) | 94 (93.1) | 101 | 7 (6.8) | 96 (93.2) | 103 | .97 | 1.02 | 0.34–3.02 | 1.0 |
| Socialized into extremism by family/peers by age 13[e] | No / Yes / Unknown | 8 (9.9) | 73 (90.1) | 81 | 6 (8.1) | 68 (91.9) | 74 | .70 | 1.24 | 0.41–3.76 | 6.0 |
| Divorced parents | No / Yes / Unknown | 50 (50.5) | 49 (49.5) | 99 | 47 (48.5) | 50 (51.5) | 97 | .77 | 1.09 | 0.62–1.90 | 4.9[f] |
| Parents involved in crime | No / Yes / Unknown | 15 (17.9) | 69 (82.1) | 84 | 9 (11.3) | 71 (88.8) | 80 | .23 | 1.72 | 0.70–4.18 | 20.4 |
| Death of parent / guardian | No / Yes / Unknown | 10 (10.0) | 90 (90.0) | 100 | 9 (9.2) | 89 (90.8) | 98 | .85 | 1.09 | 0.43–2.83 | 3.9[f] |
| *Individual level of analysis: radicalization dynamics* | | | | | | | | | | | |
| Grievance before radicalization | No / Yes / Unknown | 50 (53.8) | 43 (46.2) | 93 | 47 (48.5) | 50 (51.5) | 97 | .46 | 1.10 | 0.43–2.83 | 7.8 |
| Grievance during radicalization | No / Yes / Unknown | 102 (99.0) | 1 (1.0) | 103 | 95 (95.0) | 5 (5.0) | 100 | .09 | 5.36 | 0.61–46.79 | 1.5 |
| Extremist role model before radicalization | No / Yes / Unknown | 96 (97.0) | 3 (3.0) | 99 | 87 (86.1) | 14 (13.9) | 101 | .01 | 5.15 | 1.43–18.52 | 2.9 |
| Duty of care before radicalization | No / Yes / Unknown | 92 (89.3) | 11 (10.7) | 103 | 84 (82.4) | 18 (17.6) | 102 | .15 | 1.79 | 0.80–4.01 | 0.0 |
| Duty of care during radicalization | No / Yes / Unknown | 77 (74.8) | 26 (25.2) | 103 | 61 (59.8) | 41 (40.2) | 102 | .02 | 1.99 | 1.10–3.61 | 0.0 |
| Viewpoint diversity before radicalization[g] | No / Yes / Unknown | 100 (97.1) | 3 (2.9) | 103 | 103 (100.0) | 0 (0.0) | 103 | .25 | 0.14 | 0.01–2.72 | 0.0 |
| Viewpoint diversity during radicalization | No / Yes / Unknown | 89 (86.4) | 14 (13.6) | 103 | 97 (94.2) | 6 (5.8) | 103 | .06 | 0.39 | 0.14–1.07 | 0.0 |

(*Continued*)

**Table 4.** (Continued)

| Variable | Operationalization | Involved in terrorist violence | | | | Non-involved in terrorist violence | | | | | p | OR | 95% CI | % Missing data |
|---|---|---|---|---|---|---|---|---|---|---|---|---|---|---|
| | | Yes | | No | | Sum | Yes | | No | | Sum | | | | |
| | | -2 | -1 | 0 | 1 | 87 | -2 | -1 | 0 | 1 | 96 | .00 | 0.16 | 0.08–0.33 | 1.6 |
| Viewpoint diversity direction during radicalization[h] | -2 Ceased; -1 Decrease; 0 Stability; 1 Increase | 29 (33.3) | 46 (52.9) | 10 (11.5) | 2 (2.3) | | 12 (12.5) | 36 (37.5) | 29 (30.2) | 19 (19.8) | | | | | |

*Notes*. N = 206. Entries in the table are counts, with percentages out of column totals in parentheses. Odds ratios were calculated according to Altman [184] and standardized mean difference (d) using [185]. Significant findings are reported only where they contrast with non-significant pre- or post-radicalization onset measurements.

[a] Jihadist sub-group only (*n* = 103)

[b] For those enrolled in education (*n* = 129)

[c] For those in a relationship (*n* = 111)

[d] Higher scores indicate lower social isolation. A non-parametric equivalent (i.e., the Mann Whitney U test) was used as the data were negatively skewed and assumptions of homogeneity not met.

[e] Comparison excludes those exposed to "radical" views (*n* = 38). Given the unlikeliness of this being missed by investigators, mental health professional or journalists, unknowns were counted as "no's" for this variable.

[f] Those with missing data for this variable were significantly older than valid cases.

[h] Cell counts below minimum expected count. Significance level is interpreted using Fisher's Exact Test.

[g] For those who experienced viewpoint diversity during radicalization (*n* = 186).

homegrown extremist who identifies with the combatants or their cause will engage in terrorist violence. It may be the case that overseas conflicts can catalyze involvement in terrorist violence in other ways, such as by rousing strong emotions, providing extremist "role models" whose ideas or actions inspire emulation, or offering practical examples of the claims that extremist ideologies make about the necessity for armed resistance. Moreover, overseas conflicts involving groups that a radicalized individual identifies with can be seen as a geopolitical source of grievance. This brings to mind that, while prevalence findings can initially support a causality hypothesis based on association strength and consistency [187], non-linear analyses may identify more complex patterns, particularly with the inclusion of known *mediators* of grievances and political violence, such as political opportunity structure [188] and perceived anger [189].

We are also cognizant that our binary measurement is methodologically limited and may obscure *the degree* to which such conflicts influence radicalization trajectories. As Braithwaite and Chu [42] have shown, the likelihood that conflicts involving foreign fighters will yield domestic terrorism is subject to various nuances, such as whether the fighting is ongoing and which side achieves victory. The foreign-fighter phenomenon that followed the rise of the so-called Islamic State (IS) in the mid-2010s, and which saw thousands of individuals from across the globe travel to Syria and Iraq to bolster the group's ranks, is a particularly interesting example of conflict spillover. Politicians and policymakers have evinced concern that the paramilitary skills and experience gained by foreign fighters could make them markedly successful in organizing or carrying out terrorist attacks upon return home [190, 191]. Nonetheless, with less than 3% missing data and no evidence of confounding, we believe our null finding on the presence of such a conflict increasing the likelihood that domestic radicalization trajectories will lead to a terrorist attack, to be noteworthy. As such, we join a broader literature skeptical of presuming a notable, or straightforward, association between the presence of overseas conflicts that attract foreign fighters, and a domestic terrorist threat arising from individuals or groups who identify with the foreign fighters' cause or ideology [192–194].

We are also reminded of terrorism studies' recency-bias, in which various incentives produce a tendency among scholars to focus on the "latest threat" [28]. This dynamic imputes a degree of a-historicism in which the most recent iteration of a phenomenon is seen as representative of its characteristics more broadly. Certainly, the rise of IS heralded both unprecedented foreign-fighter streams as well as domestic terrorist attacks, some of which were committed by returned foreign fighters. However, other instances of overseas conflicts drawing domestic extremists, such as the neo-Nazis who fought in the violent dissolution of Yugoslavia in the 1990s, the thousands who traveled to defend Republican Spain in the 1930s and, indeed, those who more recently traveled to Syria and Iraq to *oppose* jihadist forces, lacked a notable homegrown terrorism component [195, 196]. History does not contain ready-made answers for pressing issues, but it can provide worthwhile context for both the study and prevention of terrorism.

## Individual-level findings

The majority of our variables fell under the individual level of analysis. After determining that data pertaining to these variables were missing at random, we observed several null findings, some of which reflect noteworthy debates in the field. To begin with, we believe that our findings align with a central premise of the criminological theory of "social control", which postulates that individuals with pro-social ties to society, such as those gained through employment, educational commitments, or being in a relationship, are less likely to engage in criminal or delinquent behavior [85]. Like other recent applications of this theory to the study of terrorism [57, 84], we found that radicalized individuals with such pro-social ties were less likely to engage in terrorist violence (though we hasten to add that, in many cases, they still engaged in other illegal, and oftentimes violent, activities in furtherance of their extremist convictions). However, with regard to several of these pro-social ties, we observed that their ability to protect against radicalization processes yielding involvement in terrorist violence appears to be temporally delineated. That is, they were only associated with radicalization processes *not* leading to terrorist violence when they were present *after* radicalization onset, but not *before* these processes got underway.

For instance, our results indicate that being employed before radicalization onset was not significantly associated with the outcome, but it was associated with non-involvement in terrorist violence *during* radicalization. In other words, for those who were employed during radicalization, their convictions were less likely to manifest in an actual terrorist attack. With regard to education, the *type* or *level* of education (e.g., primary, secondary, etc.) completed prior to radicalization was not significantly associated with future radicalization process outcomes. This tentatively suggests that the level or extent of education completed prior to education is unlikely to indicate whether radicalization will lead to involvement in terrorist violence. This was also the case for enrolment in any type of education post radicalization-onset, which was not associated with the outcome. However, for radicalized individuals enrolled in education, a gradual lessening, or even abandonment, of those responsibilities was significantly associated with involvement in terrorist violence.

These temporal disparities lead us to a number of preliminary conclusions. First of all, they bring to mind the distinction between "promotive" and "protective" factors. The former are variables whose presence is associated with a lower probability of future offending, whereas the latter interact with risk factors to limit or nullify their influence [142]. Our results suggest that employment may function in a protective, but not a promotive, manner. Second, regarding education, our results suggest that merely noting whether a radicalized individual had educational commitments at some point during their radicalized period in life (i.e., coding this

item in a purely binary fashion) is unlikely to accurately capture the protective potential of this variable. However, observing that abandonment of education was significantly associated with involvement in terrorist violence, an interesting avenue of enquiry may be to examine the factors that intersect when an individual is *in* education.

These are not the only variables for which we noted temporal shifts in significance levels. The pattern of *increasing* statistical significance when the variable is observed after radicalization onset is also seen with regard to substance abuse, being in a relationship, becoming a parent, being homeless, degree of social isolation, having a grievance, and having a duty of care (e.g., to children, elderly parents). Interestingly, the *reverse* pattern of *decreasing* statistical significance was noted in relation to various components of non-terrorism related criminal behavior. Whereas violent and non-violent criminal offences committed before radicalization onset show a statistically significant association with future involvement in terrorist violence, when committed *during* radicalization, the association is no longer present.

We hypothesize that biographical stressors, such as substance abuse, unemployment, or homelessness primarily become risk factors for radicalization leading to terrorist violence once they are interpreted through the lens of an extremist ideology. Radicalization outcomes in Western countries may be less about the objective conditions to which an individual has been exposed, and more about the cognitive changes that allow an individual to see their situation in a different, highly politicized light; not as the unfortunate results of happenstance, personal shortcomings, or impersonal structural inequalities, but as injustices perpetrated by a corrupt socio-political system that cannot and need not be accepted any longer.

As Della Porta [38 p. 136] has aptly observed, "conversion to violence requires a specific redefinition of reality". The cognitive change that is inherent in any radicalization process may make pre-existing grievances suddenly salient, rather than those grievances or biographical stressors "predicting" whether an individual will radicalize to involvement in terrorist violence. We believe that this argument is supported by our observation that those who became involved in terrorist violence were coded as having significantly higher levels of social isolation after radicalization onset only. It is precisely in settings where individuals lose old social ties (e.g., because they moved to a new city, ended a relationship, or quit a job) that their openness to new social networks and time to explore new ideas increases [197]. This is not to say that personal characteristics such as low self-control [198, 199], or biographical aspects such as adverse childhood experiences [200], are not relevant for understanding radicalization onset and outcome. We do, however, believe that our findings urge caution towards understanding radicalization as *principally* happening to "at risk" individuals.

Our observation that violent and non-violent criminal antecedents are significantly associated with radicalization leading to terrorist violence, but lose this association if committed *during* radicalization, constitutes another key finding. We believe that this relationship emphasizes the overlap between criminal and terrorist social milieus, the so-called "crime-terror nexus". Essentially, criminals are both especially sought-after by extremist groups and movements for their particular skillset, and themselves more likely to seek redemption or increased status by placing their criminal expertise at the disposal of their newfound extremist convictions [201]. Criminal antecedents appear to increase individual susceptibility to radicalization leading to terrorist violence by increasing the likelihood of participation in extremist social milieus, losing their relevance once radicalization is initiated. Interestingly, pre-radicalization involvement in a criminal gang does not match this pattern. While we have no theoretically-supported explanation for this observation, it may be that the strong sense of belonging–and potentially severe penalties against those who try to leave them behind–may make a transition from criminal gang membership to terrorism more difficult than it is for "unaffiliated" or "independent" criminals. In general, however, we argue that this set of findings speaks to the

importance of considering temporality when studying involvement in terrorism, and supports calls for more emphasis on life-course perspectives in this context [143].

Scholars have long been interested in whether terrorism is a product of mental illness [89]. While we detail our statistically significant findings related to sub-clinical and professionally-diagnosed mental illnesses elsewhere [83], here we report that we did not find neurodevelopmental issues (i.e., conditions such as attention-deficit disorder that are distinct from mental illnesses) to be associated with radicalization leading to terrorist violence. These findings support a broader skeptical position on the extent to which clinical diagnoses confer risk of involvement in terrorist violence [24]. Turning to other biographical vulnerabilities, we found that, although substance abuse and homelessness both increased in statistical significance after radicalization onset, they were not associated with involvement in terrorist violence pre-radicalization. Here, we argue that despite emerging as salient predictors of radicalization itself (when compared with the broader population [202, 203]), the relevance of these stressors in predicting radicalization *outcomes* appears limited.

While the presence of a romantic relationship was not associated with the outcome prior to radicalization, this variable was associated with non-involvement in terrorist violence post radicalization-onset. A similar dynamic was observed for an individual becoming a mother or father, or for having a duty of care (e.g., to elderly parents). These findings further underline the importance of the temporal dimension of exposure to risk and protective factors. As we also noted in relation to employment, our results suggest that a variety of pro-social ties exert a protective but not a promotive effect when it comes to the likelihood that radicalization processes will lead to involvement in terrorist violence. This is not only interesting from a research perspective but suggests to counterterrorism policymakers and practitioners that the *timing* of prevention-oriented interventions is as important as their contents. For instance, while emphasizing someone's duty of care towards elderly parents or children when they are in the pre-radicalization "at risk" phase may be ineffective as a means of preventing terrorist attacks, it appears considerably more fruitful when considered *after* said individual has begun to radicalize.

Indeed, this notion may be rooted in the broader mechanisms of viewpoint diversity and pro-social ties. The existing literature suggests that radicalizing individuals who maintain pro-social interpersonal ties, or retain viewpoint diversity in their social networks, appear less likely to endorse or engage in terrorist violence [138, 204, 205]. While we found viewpoint diversity as such to fall (just) outside of statistical significance (both before and after radicalization onset), we did find a strong association between *maintaining* or *increasing* such diversity after radicalization and a lower likelihood of turning to terrorism [83]. This offers another example of the relevance of exploring such variables in a dynamic fashion that can account for change over time, rather than as static constructs that are either present or absent.

One particular set of non-significant findings concerns the family and upbringing of individuals who will radicalize later in life. While parental divorce and death of a parent or guardian have been highlighted as risk factors for delinquency as well as radicalization specifically [200, 206], our results suggest that their relevance for whether or not radicalization processes will lead to involvement in terrorist violence, is limited. This is not necessarily an insurmountable contradiction. It may be the case that family and upbringing-related stressors are best understood in relation to *other* risk and protective factors or, alternatively, may be primarily relevant for understanding radicalization onset, rather than outcomes.

Research shows that parental political preferences play a significant role in shaping those of their children [207]. However, we found that having family members involved in extremism or terrorism, being socialized into extremist views (as opposed to "radical" ones, see Research Design) by family members or peers, or having an extremist role model prior to radicalization,

were not significantly associated with involvement in terrorist violence. Here, we tentatively hypothesize that early exposure to extremist views through family and peers who endorse extremist views may facilitate their adoption through social learning [56], essentially making it more likely that *cognitive* radicalization will occur. However, when it comes to putting such convictions into practice, observations of the high personal costs (e.g., death, imprisonment) and negative impact on family life may render the actual adoption of terrorist violence as a means for pursuing sociopolitical change decidedly *un*attractive.

A final set of findings can be more succinctly discussed. While converts to Islam have drawn scrutiny for being overrepresented among jihadist extremists [208], we found no significant association between religious conversion and involvement in terrorist violence. Arguably, noteworthy cases of converts who became jihadists and conducted terrorist attacks have created a skewed impression of conversion to Islam as conferring risk of radicalization to terrorist violence. It is important to note, however, that, here and elsewhere, our focus on radicalization *outcomes* means we cannot speak to its salience in relation to radicalization *onset*. Finally, some research suggests an association between the length of an individual's involvement in extremist or terrorist groups or movements and the likelihood they will carry out an attack [209, 210]. By contrast, our analyses failed to reveal a statistically significant difference in time spent radicalized between those who became involved in terrorist violence and those who remained non-involved. Our slightly different conceptualizations of "time involved" may play a role here, but we nonetheless hypothesize that this temporal dimension may not be particularly relevant for understanding the likelihood that a radicalized individual will become involved in terrorist violence.

## Conclusion

Although quantifying evidence is an inherent objective of empirical science, the preference for significant findings in publication culture jeopardizes this tenet by incentivizing researchers to produce significant results, thus increasing the likelihood of false-positives [211]. We argue that reporting statistically non-significant associations with involvement in terrorist violence may reveal as much as significant, bivariate findings. Our analysis suggests that risk and protective factors for radicalization are not static constructs, and that the influence of variables such as employment, educational enrollment and social isolation is tied primarily to their development over time, rather than their presence or absence at any given moment. Numerous variables gained or lost statistical significance when pre- and post-radicalization onset measurements were taken. For academics, this suggests the relevance of taking a temporal perspective into account when seeking to explain involvement in terrorist violence. For policymakers and practitioners working to detect and prevent terrorist attacks, the successful deployment of interventions aimed at reducing the likelihood of such events may depend, in large part, on whether this occurs when an individual is at risk of, or has already begun to, radicalize.

That the salience of numerous variables for understanding involvement in terrorist violence increased after radicalization onset leads to a second key conclusion. Namely, that an individual's vulnerability to radicalization leading to involvement in terrorist violence may be less about the objective conditions to which they are exposed, or the constellation of stressors they have experienced, than about their learning to see them in a different light. Just as radicalization brings individuals to reinterpret the world around them, often in starkly Manichean terms, they may also revisit disappointments and grievances in their personal lives and come to see those, too, as resulting from the nefarious work of their recently identified adversaries. Our findings suggest that, rather than seeking explanations for radicalization to terrorist violence as rooted solely in individual experiences, vulnerability or even pathology, it is the

radicalization process itself that forms the primary risk factor. It seems particularly worthwhile to expand the search for causes of radicalization and terrorism by more specifically considering the circumstances in which radicalization occurs.

At the same time, it must be kept in mind that the variables reported here are statistically non-significant in the context of radicalization process *outcomes* only. As many were informed by radicalization theories more broadly, it remains a possibility that some may predict radicalization *onset*. Furthermore, while we see the broad geographical scope, inclusion of right-wing as well as jihadist extremists, and group-based as well as lone-actor terrorists as strengths of our study by producing insights with broad-reaching rather than specific relevance, disaggregation along these lines may reveal notable shifts in patterns of (non-) significance. Just as mental illness rates are considerably more pronounced among certain perpetrator types [24], it is likely that a closer look at other sub-groups of radicalized individuals will reveal further variations.

Research on terrorism appears to lag behind other fields in utilizing replication studies to test the validity of extant findings. We believe to have demonstrated the value of considering statistically non-significant findings for advancing research on involvement in terrorist violence and hope that our findings can help place greater, field-wide emphasis on reproducibility. Moreover, far from simply signifying that a particular theory or concept is irrelevant for understanding terrorism, we argue that non-significant findings can help us clarify where theoretical assumptions lack depth, as well as the influence of temporality. Clearly, there is considerable potential to further consider how statistically non-relevant findings can help us understand involvement in terrorist violence. Sometimes, it's what you don't find that matters.

## Supporting information

**S1 Table. Missing data procedures for all variables.**
(PDF)

**S1 File. Sample interview guide.**
(PDF)

**S2 File. Interviewee consent form (English).**
(PDF)

**S3 File. NITV dataset codebook.**
(PDF)

## Acknowledgments

The authors thank all interviewees for their time and trust, the Dutch Public Prosecution Service for granting access to privileged data on numerous Dutch terrorism cases, and the reviewers and editor for their insightful and constructive feedback.

## Author Contributions

**Conceptualization:** Bart Schuurman.

**Data curation:** Bart Schuurman.

**Formal analysis:** Sarah L. Carthy.

**Funding acquisition:** Bart Schuurman.

**Investigation:** Bart Schuurman, Sarah L. Carthy.

**Methodology:** Bart Schuurman, Sarah L. Carthy.

**Project administration:** Bart Schuurman.

**Resources:** Bart Schuurman.

**Supervision:** Bart Schuurman.

**Validation:** Bart Schuurman, Sarah L. Carthy.

**Visualization:** Sarah L. Carthy.

**Writing – original draft:** Bart Schuurman, Sarah L. Carthy.

**Writing – review & editing:** Bart Schuurman, Sarah L. Carthy.

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
