## [Decision Letter · Decision Letter 0]

9 Feb 2023

PONE-D-22-27607Contextualizing involvement in terrorist violence by considering non-significant findings: Why null results matterPLOS ONE

Dear Dr. Schuurman

Many thanks for submitting your manuscript “Contextualizing involvement in terrorist violence by considering non-significant findings: Why null results matter” (with Sarah Carthy) to PLOS ONE. The review process is now complete as I have received comments and recommendations from two reviewers. I was very happy to be offered the help of two highly renowned experts, in whose advice I trust a lot and whom I consider ideal consultants for the evaluation of this submission. I have also read the manuscript carefully myself. Please find the two sets of reviewer comments appended to this letter.

My expert advisors are quite content with your manuscript. To quote from Reviewer 1, “The manuscript is beautifully written, well constructed, and definitely worthy of publication.” Reviewer 2 also feels that “This manuscript is interesting. It addresses an interesting and important question ….” 

Nevertheless, Reviewer 2 and myself each have some concerns. Because I believe these concerns can be addressed, I am offering you an opportunity to revise and resubmit the manuscript. Reviewer 2 makes a number of excellent points and I will not re-iterate all of them here. Instead, I will add several of my own.

I appreciate the attempt to drawn more attention to non-significant results but as Reveiwer 2, I am afraid these results are not that novel. Also, by mainly looking at linear relationships between various factors and terrorist violence it may reduce our understanding and the complexity of terrorist violence.

First, we already know that structural/root cause factors are not reliable predictors of terrorist violence. To the best of my knowledge support for this linear relationship is limited (e.g., Enders & Hoover, 2012; Krueger & Malečková, 2003; Abadie, 2006; Krueger et al., 2003; Piazza, 2006). In contrast, research suggests the opposite (Bhui et al., 2014). For example, many scholars have proposed that root cause/structural factors are probable necessary conditions, but not sufficient conditions to explain the phenomenon of terrorism violence (see, Sageman, 2004; Kruglanski & Fishman, 2009; Sageman, 2014).

I am also concerned with your claim that the null finding lends support to the insufficiency of grievances. I agree that objective grievances have limited merit explaining terrorist violence because subjective (psychological) feelings may not necessarily mirror objective conditions. But we know from social psychological research that it is not the perception of injustice (cognitive component) but feelings (affective component) that matter most for collective action (Starmans et al., 2017; Smith et al., 2012). Related to this, my own work (Obaidi et al., 2018; political psychology) did not find a direct effect of perceived injustice on violent extremism but the relation was mediated by perceived anger.

Related to the above I was wondering if the various data set/studies reported any mediation or moderation analysis. For example, it has been proposed by many that Muslim identification is important for understanding terrorist violence but empirical studies often find no direct relation between Muslim identification and terrorist violence, but the effect is mediated by group efficacy or/and intergroup emotions.

I would like to see some information regarding the different measures. For example, I am curious to know how grievance is measured in different data sets. The predictive power of grievance is closely related to how we measure grievance (see Smith et al., 2011). As Pettigrew (2016) puts it: “(a) People first make cognitive comparisons, (b) they next make cognitive appraisals that they or their ingroup are disadvantaged, and finally (c) these disadvantages are seen as unfair and arouse angry resentment. If any of these three requirements is missing, grievance is not operating” (p.9).

It also important to acknowledge the use of secondary data and it shortcoming, which has been a major issue in research on terrorism. Moreover, more than half of the cases consisted of semi-structured interviews with former 207 extremists and terrorists (N = 37), autobiographical materials written by extremists and terrorists (N = 56) and case files provided by the Dutch Public Prosecution Service (N = 19). Autobiographical materials written by former terrorists have the potential for romanticizing the self or of engaging in self-indulgence. We already know that social desirability bias, self-presentation, introspection and objectivity are of huge concern in this kind of data.

I think the reader may also want to see the codebooks as part of the SOM.

As Reveiwer 2, I don’t really understand the power rational. Please, address this.

Finally, in page 21 you write “… we did not find neurodevelopmental issues (i.e., conditions such as attention-deficit disorder that are distinct from mental illnesses) to be associated with radicalization leading to terrorist violence.” I wonder how these findings align with research showing that 6% of foreign fighters had diagnosable disorders such as psychotic, narcissistic, attention-deficit/hyperactivity disorder, attention-deficit disorder, PTSD, schizophrenia and autism spectrum (Weenink, 2015).

I invite you to revise and resubmit your manuscript in response to my comments and those of the Reviewers. If you choose to undertake a revision, please include a response letter that indicates how you responded (or why you chose not to respond) to each comment.

Please submit your revised manuscript within 90 days from receipt of this letter. If you will need more time than this to complete your revisions, please reply to this message or contact the journal office at plosone@plos.org. Please include the following items when submitting your revised manuscript:A rebuttal letter that responds to each point raised by the academic editor and reviewer(s). You should upload this letter as a separate file labeled 'Response to Reviewers'.A marked-up copy of your manuscript that highlights changes made to the original version. You should upload this as a separate file labeled 'Revised Manuscript with Track Changes'.An unmarked version of your revised paper without tracked changes. You should upload this as a separate file labeled 'Manuscript'.

I look forward to receiving the revised manuscript and thank you for considering PLOS ONE as an outlet for your research.

Kind regards,

Milan Obaidi

Academic Editor

PLOS ONE

Journal Requirements:

2. We noted in your submission details that a portion of your manuscript may have been presented or published elsewhere. Please clarify whether this publication was peer-reviewed and formally published. If this work was previously peer-reviewed and published, in the cover letter please provide the reason that this work does not constitute dual publication and should be included in the current manuscript.

Reviewers' comments:

Reviewer's Responses to Questions

**Comments to the Author**

1. Is the manuscript technically sound, and do the data support the conclusions?

Reviewer #1: Yes

Reviewer #2: Yes

2. Has the statistical analysis been performed appropriately and rigorously? 

Reviewer #1: Yes

Reviewer #2: Yes

3. Have the authors made all data underlying the findings in their manuscript fully available?

Reviewer #1: Yes

Reviewer #2: Yes

4. Is the manuscript presented in an intelligible fashion and written in standard English?

Reviewer #1: Yes

Reviewer #2: Yes

5. Review Comments to the Author

Reviewer #1: This manuscript discusses several structural and individual-level explanatory variables which failed to emerge as significant predictors of involvement in terrorist violence in a dataset (N = 206) of right-wing and jihadist extremists active in Europe and North America. The manuscript illustrates how some variables emerge as salient predictors of involvement

in terrorist violence during a lifespan, but not others.

This manuscript interrogates and interesting and important phenomena. Non-significant results are often as, if not more, important that statistically significant results. However, in this field, there is a distinct lack of interest in such reporting. The manuscript is beautifully written, well constructed, and definitely worthy of publication.

However, prior to publication, Table 2 should be split into different tables to reflect the different forms of analysis conducted. This will help in the interpretation of both the results and the areas of discussion.

Reviewer #2: This manuscript is interesting. It addresses an interesting and important question of what we can learn from insignificant findings regarding predictors of involvement in terrorist violence. The manuscript reviews a number of individual level, situational and systemic factors that extant literature highlights as candidates for predicting involvement in terrorist violence and use a data set of Islamist and far-right extremist (N=206), who all radicalized in terms of ideology and acceptance of violence as a political mean, but only half ended up engaging in terrorist violence, to test the significance of these potential predictors. The manuscript argues that we can learn from the insignificant results, especially when a given variable turns out to be insignificant in terms of predicting involvement in terrorist violence at the outset of radicalization, but becomes a significant predictor during radicalization. As such, the article argues that while some static factors fail to predict terrorist involvement, treating the factors over time (during radicalization) produces different results. Thus, radicalization may have less to do with individual pre-disposing factors and more to do with dynamic development of e.g. social ties over time. These are important points often overlooked in the literature. However, the manuscript also suffers from substantial weaknesses:

1) While the conclusion about the need to look at risk factors of involvement in terrorist violence in a more dynamic/temporal way than allowed by simple regression models is important, it is hardly new. Well, it might be a new insight reached from a variance-based outset/methodology. However, research on involvement in terrorist violence departing from more case-based and relational perspectives have argued along these lines for years (e.g. Della Porta 1992; McAdam 1986; McAdam and Paulsen 1993; Moskalenko and McCauley 2009; Passy 2001; Sageman 2004; Snow, Zurcher, and Ekland-Olson 1980; Wiktorowicz 2005; van Stekelenburg and Klandermans 2013; for an overview see Malthaner 2017). The manuscript needs to address this literature and show how the findings highlighted in the manuscript aligns with the arguments of this relevant literature in order to establish a contribution.

2) There is a tendency to talk about different variables’ ‘effect’ on the outcome (engagement/non engagement in terrorist violence) and observed differences as ‘effect sizes’, suggesting a causal relationship between factors. As all analysis on which the manuscript rests are purely correlational this is unfortunate and misleading. The manuscript should take greater care in formulations here and needs to explicitly address the difference between causal and purely correlational relationships.

3) The individual matching of cases is not sufficiently described. Did this matching happen within e.g. the ‘islamist’ and ‘far-right’ samples or across. This process of matching needs to much better described.

4) It is unclear why ‘Islamist’ and ‘far-right’ cases are lumped together, despite these being two main types of radicalization in the West. Likewise, why lump group-based and lone actor terrorists together? This lumping together will increase sample bias and the likelihood of non-significant results. The consequences of these choices for the findings and conclusions drawn needs to be better discussed.

5) The issue of statistical power to detect statistically significant differences in the analysis needs to be much more explicitly addressed. What does the N=206 mean for detecting such differences and what would happen to our chances of detecting differences if the ‘lumping problem’ was solved by splitting the data in two - an islamist and a far-right dataset of N=103 each? How is the calculation of X2 sensitive the size of N?

6) This reviewer misses a more detailed discussion of the problem of replication of findings within terrorism research; what are the causes of this? Data problems? Design problems? Publication biases?

7) The manuscript would benefit from a discussion of to what degree the relevance of looking at non-significant findings/null findings is particular to the terrorism research field. How does this relevance change e.g. when turning to a more advanced research field where the quality and extent of empirical studies is higher?

Based on this and providing an overall assessment of the manuscript, I recommend inviting the authors to revise and resubmit their manuscript after major revisions.

Smaller issues:

- line 119, should ‘lower’ not be ‘increase’?

- table 2 is difficult to read – can this be simplified somehow?

6. PLOS authors have the option to publish the peer review history of their article (what does this mean?). If published, this will include your full peer review and any attached files.

Reviewer #1: No

Reviewer #2: No

---

## [Author Response · Author response to Decision Letter 0]

5 Jul 2023

Dear Professor Obaidi,

Thank you very much for providing us with this opportunity to improve our manuscript by incorporating reviewer feedback. We appreciate the time that the reviewers and yourself have taken to critique our work, as this has provided us with numerous opportunities to strengthen our paper. Thank you also for your patience and flexibility in allowing us to extend the submission deadline for our revisions, which we were unfortunately forced to do due to personal circumstances affecting the first author.

Please find below an overview of the reviewer’s comments and the changes we made in response to them. We have also incorporated a small number of tweaks to the phrasing of several sentences, none of which change anything about the substance of the paper. For ease of reference, we have uploaded two versions of our manuscript; one anonymized version with track-changes highlighted and one non-anonymized “clean” version. 

As requested, we would also like to emphasize that this manuscript makes a standalone contribution, even though it is part of a larger, multi-year project on (non-) involvement in terrorist violence that we have been working on. No portions of the current manuscript have been published elsewhere. Three other articles based on this project have been published in the past 12 months, and two others are currently under review. We have attached copies of all five manuscripts so that the reviewers and yourself may confirm that there is no undue overlap between those papers and the current manuscript, nor that we are replicating previously published findings.

We look forward to receiving your appraisal of our revised manuscript.

Yours sincerely,

Bart Schuurman & Sarah L. Carthy

Leiden University

The Netherlands

---

E0: My expert advisors are quite content with your manuscript. To quote from Reviewer 1, “The manuscript is beautifully written, well-constructed, and definitely worthy of publication.” Reviewer 2 also feels that “This manuscript is interesting. It addresses an interesting and important question ....” 

Dear Dr. Obaidi, we appreciate the work that you and the reviewers have put into assessing our manuscript and are happy to read that it was found praiseworthy.

Editorial feedback

E1: I appreciate the attempt to drawn more attention to non-significant results but as Reviewer 2, I am afraid these results are not that novel. Also, by mainly looking at linear relationships between various factors and terrorist violence it may reduce our understanding and the complexity of terrorist violence. 

We appreciate this point about the limitations of taking a linear approach and regret that we did not demonstrate our familiarity with the extensive evidence base illustrating this very point This point has now been emphasized in our paper, including the abstract and introduction. Please see our response to E2 for further details.

Regarding the question of novelty, we have a) considerably expanded our literature review to acknowledge other relevant work on this subject, and b) explicitly stated our contribution to this broader debate. We do so in the newly added “Contribution to the literature” section (pp. 16-19). There, we argue that the novelty of our work lies particularly in a) specifically focusing on non-significant findings, b) explicitly distinguishing between risk and protective effects, and c) using a temporal perspective to assess how the significance level of (some of) our predictors is liable to change during the transition from pre- to post-radicalization onset. We believe this allows us to contextualize predictors of involvement in terrorism in a unique fashion that adds a novel perspective to the literature, strengthened by a sizeable dataset, our use of unique primary data, and a multi-country sample. We have also made sure to more clearly state the purpose of our paper in key sections such as the abstract and introduction.

E2: First, we already know that structural/root cause factors are not reliable predictors of terrorist violence. To the best of my knowledge support for this linear relationship is limited (e.g., Enders & Hoover, 2012; Krueger & Malečková, 2003; Abadie, 2006; Krueger et al., 2003; Piazza, 2006). In contrast, research suggests the opposite (Bhui et al., 2014). For example, many scholars have proposed that root cause/structural factors are probable necessary conditions, but not sufficient conditions to explain the phenomenon of terrorism violence (see, Sageman, 2004; Kruglanski & Fishman, 2009; Sageman, 2014). 

We fully agree with this point and we have amended our literature review section to clarify two issues. First, we have expanded the discussion of structural-level factors in particular to more accurately reflect some of the major findings and debates in this area (using many of the references you have helpfully provided here). Secondly, we use this expanded literature review to explicitly clarify that we did not seek to test old-fashioned notions of structural-level variables, such as poverty at a societal level potentially being a “root cause” of terrorism. Instead, we see structural-level factors as shaping the broader context in which terrorism may, or may not, emerge. We re-emphasize this point when our literature review moves on to variables at the movement and group level of analysis. As noted in our response to E1, we have also touched upon the limitations of taking a linear approach by including some additional detail in the introduction. 

In other words, we acknowledge that research has already clearly established that structural-level factors such as poverty are insufficient as explanations for involvement in terrorism. Our paper does not attempt to retread this well-covered terrain. Instead, our project looked at a select number of other structural-level factors while focusing the brunt of its attention on the more strongly supported group-, movement-, and individual-level factors that the literature has put forward as explanations for involvement in terrorist violence. 

E3: I am also concerned with your claim that the null finding lends support to the insufficiency of grievances. I agree that objective grievances have limited merit explaining terrorist violence because subjective (psychological) feelings may not necessarily mirror objective conditions. But we know from social psychological research that it is not the perception of injustice (cognitive component) but feelings (affective component) that matter most for collective action (Starmans et al., 2017; Smith et al., 2012). Related to this, my own work (Obaidi et al., 2018; political psychology) did not find a direct effect of perceived injustice on violent extremism but the relation was mediated by perceived anger. 

We agree with the editor that our claim about the insufficiency of grievances, which we made in relation to the “spillover effect” discussed on p. 27, lacked nuance. On reflection, this claim was too broad to warrant inclusion in our paper. We have removed the sentences in question and, instead, added a short reflection on the role that grievances can play that, we believe, is considerably more nuanced. We made grateful use of your own work here (Obaidi et al, 2018), as well as several other references. 

In addition, we have caveated our overall interpretation of the findings more broadly. On pp. 19-20, we now draw comparisons with the caveats necessary for interpretations of significant findings. We explain that, in the same way statistical significance is not equivalent to theoretical or clinical relevance or significance (Du Prel et al., 2009, p. 337), we do not wish to dismiss the relevance of particular variables to theoretical conceptualizations of involvement in terrorist violence. Instead, we seek to create the necessary conditions for the development of more complex, foundational theories of involvement in terrorist violence. 

E4: Related to the above I was wondering if the various data set/studies reported any mediation or moderation analysis. For example, it has been proposed by many that Muslim identification is important for understanding terrorist violence but empirical studies often find no direct relation between Muslim identification and terrorist violence, but the effect is mediated by group efficacy or/and intergroup emotions. 

This was not a question that we had posed ourselves specifically when conducting our literature review, hence it is difficult for us to now address this point in detail. That being said, we have amended our paper in several related aspects. In our revisions, we have explicitly drawn attention to a key consideration of our research, being its linear nature (pp. 20-21). We have also included references to studies which have identified mediators or moderators of our null findings, specifically Obaidi et al. (2018) as well as Dyrstad et al. (2020) who found that perceived grievances and support for political violence were mediated by perceived opportunity.

E5: I would like to see some information regarding the different measures. For example, I am curious to know how grievance is measured in different data sets. The predictive power of grievance is closely related to how we measure grievance (see Smith et al., 2011). As Pettigrew (2016) puts it: “(a) People first make cognitive comparisons, (b) they next make cognitive appraisals that they or their ingroup are disadvantaged, and finally (c) these disadvantages are seen as unfair and arouse angry resentment. If any of these three requirements is missing, grievance is not operating” (p.9). 

We have now specified on p. 14 that exposure in the current study was ascertained by the study authors and on p. 21, we have outlined our approach to ascertaining grievances. Here, we determined whether there were tangible feelings of injustice emerging from comparisons at the political, religious, or personal level(s). In other words, the presence or absence of political, religious, or personal grievances (or any combination of these) were measured as binary variables, later merged into a variable capturing the presence or absence of grievances more broadly, both before and during radicalization. We did not consider it appropriate to use psychometric measures for the coding of archival data and, as such, our measurement instrument for grievances likely failed to capture the full complexity of comparisons at the cognitive level.

That being said, our coding decisions are in line with the suggested nuance in the cited relative deprivation literature. We were cognizant, for instance, that two people in comparable contexts may perceive their situation in different ways (Smith et al., 2012, p. 220), and we did not assume that individuals in the same position would share the same feelings. As coders, we needed to observe these feelings of injustice, either through interviewee reflections, diary entries, first-hand accounts, court transcripts etc. This information has now been included on p. 21.

E6: It also important to acknowledge the use of secondary data and its shortcomings, which has been a major issue in research on terrorism. Moreover, more than half of the cases consisted of semi-structured interviews with former 207 extremists and terrorists (N = 37), autobiographical materials written by extremists and terrorists (N = 56) and case files provided by the Dutch Public Prosecution Service (N = 19). Autobiographical materials written by former terrorists have the potential for romanticizing the self or of engaging in self-indulgence. We already know that social desirability bias, self-presentation, introspection and objectivity are of huge concern in this kind of data. 

We agree with this important point and have added a new section on “Limitations” in which we specifically discuss data-related limitations. We also added a new table to clarify the types of sources used. Please see pp. 21-23 for details.

E7: I think the reader may also want to see the codebooks as part of the SOM.

We had added “S4 Supporting information. Codebook” at the end of the manuscript and uploaded the most recent version of our codebook to the PloS ONE submission system.

E8: As Reviewer 2, I don’t really understand the power rational. Please, address this. 

When referring to weakened statistical power, we are talking about the possibility that non-random distributions of missing data may produce biased estimates and, subsequently, Type I or II errors. However, upon re-reading, we see that the language we use is not always clear in this chain of thought. We have now modified this in the Abstract which now reads “After considering methodological factors such as non-random distributions of missing data” rather than “… such as statistical power”. Hopefully, this addresses the confusion.

E9: Finally, in page 21 you write “... we did not find neurodevelopmental issues (i.e., conditions such as attention-deficit disorder that are distinct from mental illnesses) to be associated with radicalization leading to terrorist violence.” I wonder how these findings align with research showing that 6% of foreign fighters had diagnosable disorders such as psychotic, narcissistic, attention- deficit/hyperactivity disorder, attention-deficit disorder, PTSD, schizophrenia and autism spectrum (Weenink, 2015). 

Many thanks for this observation. In fact, as we note in our manuscript on p. 32-33, our findings in relation to mental and neurodevelopmental disorders are in line with the broader evidence base. The cited meta-analysis of over 70 studies (including Weenink, 2015; 2019) does not support the assertion that terrorist samples are characterized by higher rates of mental health difficulties than would be expected in the general population (Sarma, Carthy & Cox, 2022). 

---

Reviewer 1

R1.1: This manuscript discusses several structural and individual-level explanatory variables which failed to emerge as significant predictors of involvement in terrorist violence in a dataset (N = 206) of right-wing and jihadist extremists active in Europe and North America. The manuscript illustrates how some variables emerge as salient predictors of involvement

in terrorist violence during a lifespan, but not others. This manuscript interrogates and interesting and important phenomena. Non-significant results are often as, if not more, important that statistically significant results. However, in this field, there is a distinct lack of interest in such reporting. The manuscript is beautifully written, well-constructed, and definitely worthy of publication. 

Dear Reviewer 1, thank you for your kind words. 

R1.2: However, prior to publication, Table 2 should be split into different tables to reflect the different forms of analysis conducted. This will help in the interpretation of both the results and the areas of discussion. 

We agree that this is a large table which can pose readability issues. In response to your query, we tried splitting it into smaller tables. To begin with, we made one table for the structural-level findings and one for the individual-level ones. However, this left us with one very small and one rather large table. We then tried to further split the individual-level table by clustering several sets of findings together. This left us with a large number of small tables. To our mind, however, neither solution proved satisfactory. Instead, we have carried out a number of changes to how the material in (what is now) Table 4 is presented in order to make it easier to read. Specifically, the font size has been increased and the different levels of analysis (i.e., structural, and individual) are now centered and shaded to differentiate them from the study variables. We hope this will be satisfactory, though we are also very happy to try another approach that the reviewer might suggest. 

Reviewer 2

This manuscript is interesting. It addresses an interesting and important question of what we can learn from insignificant findings regarding predictors of involvement in terrorist violence. The manuscript reviews a number of individual level, situational and systemic factors that extant literature highlights as candidates for predicting involvement in terrorist violence and use a data set of Islamist and far-right extremist (N=206), who all radicalized in terms of ideology and acceptance of violence as a political mean, but only half ended up engaging in terrorist violence, to test the significance of these potential predictors. The manuscript argues that we can learn from the insignificant results, especially when a given variable turns out to be insignificant in terms of predicting involvement in terrorist violence at the outset of radicalization, but becomes a significant predictor during radicalization. As such, the article argues that while some static factors fail to predict terrorist involvement, treating the factors over time (during radicalization) produces different results. Thus, radicalization may have less to do with individual pre-disposing factors and more to do with dynamic development of e.g., social ties over time. These are important points often overlooked in the literature. However, the manuscript also suffers from substantial weaknesses.

Dear Reviewer 2, we were glad to read your positive appraisal of (parts of) our manuscript and are thankful for the detailed and constructive criticism that you have offered. We have attempted to address it as best we could. Please see our responses below for details.

R2.1: While the conclusion about the need to look at risk factors of involvement in terrorist violence in a more dynamic/temporal way than allowed by simple regression models is important, it is hardly new. Well, it might be a new insight reached from a variance- based outset/methodology. However, research on involvement in terrorist violence departing from more case-based and relational perspectives have argued along these lines for years (e.g., Della Porta 1992; McAdam 1986; McAdam and Paulsen 1993; Moskalenko and McCauley 2009; Passy 2001; Sageman 2004; Snow, Zurcher, and Ekland-Olson 1980; Wiktorowicz 2005; van Stekelenburg and Klandermans 2013; for an overview see Malthaner 2017). The manuscript needs to address this literature and show how the findings highlighted in the manuscript aligns with the arguments of this relevant literature in order to establish a contribution. 

Dear reviewer, thank you for raising this important point. After studying the articles that you so helpfully outlined, we agree that they cover important subjects with clear relevance for our own manuscript. Accordingly, we have rewritten and expanded our literature review section (“Researching involvement in terrorism”, especially pp. 5-7) and added a new “Contribution to the literature” section (pp. 16-19). As a result, our manuscript now a) much more clearly acknowledges the work that others have done on related subjects (sometimes explicitly stated as such in text, in other instances through expanded references) and b) specified how our study relates and adds to this existing body of work. We have incorporated nearly all of the references you pointed us to (thank you also for the especially helpful overview found in the Malthaner, 2017 article), and brought in additional material related to recent terrorism-focused work on differentiating between violent and non-violent radicalization process outcomes.

As stated in our revised manuscript, we argue that the novelty of our work lies particularly in a) specifically focusing on non-significant findings, b) explicitly distinguishing between risk and protective effects, and c) using a temporal perspective to assess how the significance level of (some of) our predictors is liable to change during the transition from pre- to post-radicalization onset. We believe this allows us to contextualize predictors of involvement in terrorism in a unique fashion that adds a novel perspective to the literature, strengthened by a sizeable dataset, our use of unique primary data, and a multi-country sample.

R2.2: There is a tendency to talk about different variables’ ‘effect’ on the outcome (engagement/non engagement in terrorist violence) and observed differences as ‘effect sizes’, suggesting a causal relationship between factors. As all analysis on which the manuscript rests are purely correlational this is unfortunate and misleading. The manuscript should take greater care in formulations here and needs to explicitly address the difference between causal and purely correlational relationships. 

We thank Reviewer 2 for pointing this out. Despite our imprecision, both authors are cognizant of the importance of not inferring causality from these types of descriptive data, and we regret that we were inconsistent on this point throughout the manuscript. We have now made several changes throughout, replacing any reference to the “effect” of variables on the outcome to the “association” between variables and the outcome (e.g., “no effect” is replaced with “no association”. p. 16). We have also modified any other language which infers causality (e.g., the variable had “no bearing” on the outcome is replaced with the variable was “not associated with” the outcome, p. 31). 

R2.3: The individual matching of cases is not sufficiently described. Did this matching happen within e.g., the ‘Islamist’ and ‘far-right’ samples or across. This process of matching needs to much better described. 

Dear reviewer, we have now provided a more detailed description of our matching procedure in the “Sampling” section, including the newly added Table 2 which illustrates the distribution of “involved” cases and “non-involved” controls across ideological conviction (p. 14). In brief, we explain that while we sought to match each “involved” right-wing extremist with a “non-involved” right-wing extremist (and each “involved” jihadist with a “non-involved” jihadist), we did not succeed in matching the sample 1:1. We also make reference to this aspect of the study in the new “Limitations” section of the manuscript (p. 20).

R2.4: It is unclear why ‘Islamist’ and ‘far-right’ cases are lumped together, despite these being two main types of radicalization in the West. Likewise, why lump group-based and lone actor terrorists together? This lumping together will increase sample bias and the likelihood of non-significant results. The consequences of these choices for the findings and conclusions drawn needs to be better discussed. 

Thank you for raising this important question. We have addressed it at length in the “Sampling” section on pp. 11-12. We essentially argue that, while the predictors of involvement in jihadism and right-wing extremist terrorism are unlikely to be the same, they are likely to be similar enough to warrant both types of individuals being included in our dataset. Similarly, while lone actors do show distinctiveness on a number of predictors, perhaps especially so with regard to higher rates of diagnosed mental illness (Sarma et al., 2022; Corner et al., 2016), such unique attributes appear to be outweighed by similarities with their group-based counterparts (Gruenewald et al, 2013; Knight et al., 2022; Schuurman & Carthy, 2023). Hence, we argue that it makes sense to include both lone actors and group-based extremists in our work. It should also be noted that we did not specifically select cases or controls based on whether they were lone actors or operated in a group context. Instead, we specifically examined group membership as a variable of interest.

R2.5: The issue of statistical power to detect statistically significant differences in the analysis needs to be much more explicitly addressed. What does the N=206 mean for detecting such differences and what would happen to our chances of detecting differences if the ‘lumping problem’ was solved by splitting the data in two - an Islamist and a far-right dataset of N=103 each? How is the calculation of X2 sensitive the size of N? 

Many thanks for these two important points. On the second point (“What would happen if the ‘lumping problem’ was solved by splitting the data in two?”), we hope to have answered this question in response to R2.4 where we justify the use of an aggregate sample. To reiterate, we are examining radicalization as a psychological process which unfolds, comparably, across a range of extremist contexts. In this way, we join a school of scholars who examine radicalization across contexts (e.g., LaFree et al., 2018, Knight et al., 2022; Thijs et al., 2022) in a bid to contribute to a better theoretical understanding of the phenomenon more broadly. While important insights have and continue to emerge from research which disaggregates based on conviction, we believe that burgeoning, comparative research on radicalization outcomes would benefit, first, from an expansive examination before introducing disaggregation based on conviction.

We agree that the first point (“What does the N = 206 mean for detecting such differences?”), needs better clarification, and we have addressed this specific question by justifying the sample size with the following points in the ‘Sampling’ section (pp. 10-14). In line with the JBI critical appraisal checklist for studies reporting prevalence data (Moola et al., 2017), the risk of bias in relation to the following sampling factors was appraised. First, it was considered whether the sample frame was appropriate to address the target population. As we sought to examine radicalization as a psychological process across a range of extremist contexts, we defined our target population as extremists in Europe and North America who were active post 1980. In terms of the appropriateness of the sample, the entire frame was sampled to identify our “involved” cases (i.e., individuals involved in terrorist violence). This “total population sampling” approach is common in comparative research with small populations or infrequent phenomena (Etikan et al., 2016, p. 3), and is broadly appraised as allowing for better generalizability (Sharma, 2017, p. 751), and reducing the risk of bias according to the JBI checklist (Sarma et al., 2022). We hope that these considerations underpin the appropriateness of the sample size.

We are also cognizant that power and sample size estimation allow researchers to identify a sample size which achieves the desired level of precision to detect effects or associations. However, because of the total population sampling approach, power and sample size estimation were not conducted during the design phase of the current research. To allow us to better respond to your comment, a number of thresholds were calculated (in line with Sarma et al., 2022, p. 36) for variables selected from the individual level of analysis, using the following formula:

n = ((Z2)(p)(1-P))/d2

Here, n refers to the sample size, Z represents the Z statistic for the desired 0.95% confidence interval (Z = 1.96) and d represents the precision which is half the confidence interval (d = 0.5). The expected prevalence for each variable is represented by P. The selected variables were violent and non-violent crime, parental divorce or death, and mental disorder. 

From the general population comparisons cited by Schuurman & Carthy (2022), prevalence rates were identified for non-violent crime (35% - 60%) and violent crime (19% – 34%). The prevalence rates for parental divorce (13%) and parental death (8%) during childhood or adolescence were calculated from Tebeka et al., (2016). A lifetime prevalence of mental disorder (25%) was identified in Sarma et al., (2022, p. 36). With a prevalence rate of 42.5% for non-violent crime, the target sample size was n = 376. With a prevalence rate of 26.5% for violent crime, the target sample size was n = 302. For parental divorce, the target sample size was n = 173 and for parental death, the target sample size was n = 113. As reported by Sarma et al., (2022, p. 36), the target sample size to capture the prevalence of mental disorder was n = 294. 

We realize that these calculations indicate that our sample size is below where it should optimally be for several variables. However, terrorism in the Western countries that we studied is a small-N problem, and limitations related to gathering sufficient high-quality data further reduce the number of cases that can be usefully studied. Consequently, it is very rare to see samples in the 250+ range in the field of terrorism research. At 206 cases, our sample is larger, or comparable in size, to several noteworthy datasets on terrorism (Gill, Corner, et al., 2017; Gill, Silver, Horgan, & Corner, 2017; Pyrooz, LaFree, Decker, & James, 2018; Thijs et al., 2022; Knight et al., 2022). To the best of our knowledge, only Gary LaFree and colleagues have managed to collect a considerably larger sample with their PIRUS dataset of 1.400+ individuals. However, at 159 variables (the majority of which do not have accurate general population statistics), our data collection asked more of our cases and controls than other studies have done, and the relative high proportion of primary data that we managed to collect is, we believe, unique for this type of research and strengthens the reliability of what we are presenting. Our emphasis on depth of detail is further reflected in the two full years we spent on data collection. In short, a larger sample would certainly confer additional benefits, and it would be a worthwhile endeavor for future research (perhaps by including other ideological extremisms). However, we believe that our current dataset contains a robust sample size that would be difficult to substantially expand without sacrificing coding reliability and incurring a larger number of “unknowns”. 

R2.6: This reviewer misses a more detailed discussion of the problem of replication of findings within terrorism research; what are the causes of this? Data problems? Design problems? Publication biases? 

This is very interesting question, which we have addressed in two new paragraphs added to the end of the “Contribution to the literature” section (pp. 18-19). 

R2.7: The manuscript would benefit from a discussion of to what degree the relevance of looking at non-significant findings/null findings is particular to the terrorism research field. How does this relevance change e.g., when turning to a more advanced research field where the quality and extent of empirical studies is higher? 

The new paragraphs about this topic added on pp. 18-19 (see also previous comment) begin by outlining issues surrounding replicability in more established disciplines, such as psychology. We then contrast the considerable attention that replication studies have gotten in these fields, especially in recent years, with research on terrorism, where replication studies are much less common. We conclude by offering some thoughts on why this might be the case.

The introduction has also been modified to better communicate the field-specific relevance of publishing null findings. We first identify replication and publication bias as broader, scientific problems arising from a disinclination of authors to report non-significant results and, for the final two points, we contextualize these points within the field of terrorism research specifically. We initially drew attention to the “latest threat” fallacy but now, we first make an additional point related to publication bias. We argue that the reporting of null-findings helps address publication bias in systematic reviews and meta-analyses and accompanies a field-wide drive towards synthesizing terrorism and radicalization research (Carthy et al., 2020; Lum et al., 2006; Neyroud et al., 2018; Sarma et al., 2022; Wolfowicz et al., 2021). Finally, we also speak about the relevance of these types of findings to theory. As we write on p. 4, what ultimately emerges from reporting and publishing non-significant results is a mechanism for theory building through falsification. An aversion to null results underpins the endurance of “unkillable” theories (Ferguson & Heene, 2012, p. 559), whereas theory falsification, principally, allows theories to become more complex while also, hopefully, inviting theorists to develop testable theories (Earp & Trafimow, 2015, p. 2). By engaging in null-hypothesis testing, scholars are also creating the necessary conditions for more complex, foundational theories. 

R2.8: Smaller issues:

- line 119, should ‘lower’ not be ‘increase’?

- table 2 is difficult to read – can this be simplified somehow? 

Thank you for catching the mistake on line 119. That should indeed have been “increase”.

We agree that this is a large table which can pose readability issues. In response to your query, we tried splitting it into smaller tables. To begin with, we made one table for the structural-level findings and one for the individual-level ones. However, this left us with one very small and one rather large table. We then tried to further split the individual-level table by clustering several sets of findings together. This left us with a large number of small tables. To our mind, however, neither solution proved satisfactory. Instead, we have carried out a number of changes to how the material in (what is now) Table 4 is presented in order to make it easier to read. Specifically, the font size has been increased and the different levels of analysis (i.e., structural, and individual) are now centered and shaded to differentiate them from the study variables. We hope this will be satisfactory, though we are also very happy to try another approach that the reviewer might suggest.

---

## [Decision Letter · Decision Letter 1]

3 Oct 2023

Contextualizing involvement in terrorist violence by considering non-significant findings: Using null results and temporal perspectives to better understand radicalization outcomes

PONE-D-22-27607R1

Dear Dr. Schuurman,

We’re pleased to inform you that your manuscript has been judged scientifically suitable for publication and will be formally accepted for publication once it meets all outstanding technical requirements.

Kind regards,

Sylvester Chidi Chima, M.D., L.L.M, LLD,

Academic Editor

PLOS ONE

Additional Editor Comments (optional):

Reviewers' comments:

Reviewer's Responses to Questions

**Comments to the Author**

1. If the authors have adequately addressed your comments raised in a previous round of review and you feel that this manuscript is now acceptable for publication, you may indicate that here to bypass the “Comments to the Author” section, enter your conflict of interest statement in the “Confidential to Editor” section, and submit your "Accept" recommendation.

Reviewer #1: All comments have been addressed

Reviewer #2: All comments have been addressed

2. Is the manuscript technically sound, and do the data support the conclusions?

Reviewer #1: Yes

Reviewer #2: Yes

3. Has the statistical analysis been performed appropriately and rigorously? 

Reviewer #1: Yes

Reviewer #2: Yes

4. Have the authors made all data underlying the findings in their manuscript fully available?

Reviewer #1: Yes

Reviewer #2: Yes

5. Is the manuscript presented in an intelligible fashion and written in standard English?

Reviewer #1: Yes

Reviewer #2: Yes

6. Review Comments to the Author

Reviewer #1: (No Response)

Reviewer #2: I appreciate the thorough response by the authors to my concerns. The authors have dealt with these in a serious manner, which in my view have improved the quality of manuscript considerably.

7. PLOS authors have the option to publish the peer review history of their article (what does this mean?). If published, this will include your full peer review and any attached files.

Reviewer #1: No

Reviewer #2: No

---

## [Editor Report · Acceptance letter]

16 Oct 2023

PONE-D-22-27607R1 

Contextualizing involvement in terrorist violence by considering non-significant findings: Using null results and temporal perspectives to better understand radicalization outcomes 

Dear Dr. Schuurman:

I'm pleased to inform you that your manuscript has been deemed suitable for publication in PLOS ONE. Congratulations! Your manuscript is now with our production department. 

Kind regards, 

on behalf of

Professor Sylvester Chidi Chima 

Academic Editor

PLOS ONE